# Children with Hirschsprung’s Disease Report Dietary Effects on Gastrointestinal Complaints More Frequently than Controls

**DOI:** 10.3390/children10091543

**Published:** 2023-09-12

**Authors:** Lovisa Telborn, Christina Granéli, Irene Axelsson, Pernilla Stenström

**Affiliations:** 1Department of Clinical Sciences, Pediatrics, Lund University, SE-221 84 Lund, Sweden; christina.graneli@med.lu.se (C.G.); irene.axelsson@med.lu.se (I.A.); pernilla.stenstrom@med.lu.se (P.S.); 2Department of Pediatric Surgery, Skåne University Hospital, SE-221 85 Lund, Sweden

**Keywords:** dietary effects, pediatrics, Hirschsprung’s disease, gastrointestinal symptoms, daily life

## Abstract

Hirschspung’s disease (HD) is a congenital gastrointestinal (GI) disorder frequently accompanied by GI complaints. Despite the lack of evidence regarding whether diet affects GI symptoms, advice on dietary changes is common. The aim was to investigate self-reported dietary effects on GI symptoms, comparing children with HD with healthy children. This was an observational, cross-sectional, self-reported case-control study using the validated Diet and Bowel Function questionnaire. All children with HD aged 1–18 years were surgically treated during 2003–2021 at a national HD center, and their parents were invited to participate. Healthy children served as controls. The data were presented as median (range) and *n* (%). 71/85 children with HD (6 years (1–17); 76% boys) and 265/300 controls (9 years (1–18); 52% boys) participated. Dietary effects on GI symptoms were reported more frequently by children with HD than controls (55/71 [77%] vs. 137/265 [52%], *p* ≤ 0.001), as were dietary adjustments to improve GI symptoms (49/71 [69%] vs. 84/265 [32%], *p* ≤ 0.001), and social limitations due to dietary adjustments (20/48 [42%] vs. 22/121 [18%], *p* = 0.002). Of 90 food items, children with HD reported that more of the items induced GI symptoms compared to controls (7 (0–66) vs. 2 (0–34), *p* = 0.001). Diet-induced GI symptoms and dietary adjustments’ impact on daily life are reported more frequently by children with HD than controls. Moreover, the number and types of food items causing GI symptoms differ. The results indicate the need for disease-specific dietary advice to improve support for families of children with HD.

## 1. Introduction

Hirschsprung’s disease (HD) is a congenital disorder characterized by the absence of intestinal ganglion cells (aganglionosis) and streching of various lengths cranially from the anus, causing chronic bowel obstruction [1,2,3]. Despite surgery, residual gastrointestinal (GI) complaints are reported frequently [1,2,3,4]. Historically, bowel management programs for HD have recommended the use of laxative drugs and/or enemas to improve GI symptoms; however, the newest European consensus HD guidelines also include a recommendation on dietary management [1]; however, evidence is lacking. There is a paucity of knowledge concerning diet’s effects on GI symptoms in children in general [5,6,7,8], and especially of dietary effects on GI symptoms in children with HD, which are only rarely studied [9,10,11]. In our focus group study, diet was shown to play a key role in parental self-treatment of GI symptoms in children with HD, and such changes have an impact on daily life [12]. To increase knowledge on diet’s role in bowel function, we developed the Diet and Bowel Function Questionnaire [13]. In healthy children who answered the questionnaire, any GI effects as a result of diet were only evident among 52%, and the reported effects were weak [14]. In reports on children with GI disorders, diet’s effects on their symptoms were more frequent and stronger [15,16]. To clarify the role of diet in children with HD, this study aimed to investigate whether experiences of dietary effects on GI symptoms and daily life differ between children with HD and healthy children. The hypothesis was that children with HD experience dietary effects on GI symptoms more frequently and stronger.

## 2. Materials and Methods

### 2.1. Study Setting

This was an observational, cross-sectional, case-control, and patient-reported questionnaire study. Children with HD and healthy children, serving as controls, were studied. This study was conducted at the Department of Pediatric Surgery in Lund, in the south of Sweden. In 2018, the department was appointed a national referral center for HD, covering a catchment area of 5 million residents. Data collection was performed from March 2020 to March 2021.

### 2.2. Study Population

All children aged 1–18 years treated for HD were identified in the local HD register and invited to participate in this study together with their parents. Invitations and questionnaires were sent by postal mail, and a study announcement was published online by the National Hirschsprung Disease Patient Organization of Sweden [17]. Healthy children aged 1–18 years and their parents comprised the control group. They were recruited from children admitted to the hospital in order to attend single orthopedic consultations together with their siblings and the offspring of professionals working at the hospital. ‘Healthy’ was defined as being without any GI-, nephrological-, metabolic- or other disorder, being without enteral tube feeding or parenteral nutrition support, and not having any medically treated disease or allergy. Children with self-reported food allergies and/or food intolerances not requiring medication were included in this study in order to reflect a generally healthy pediatric population. The case-control ratio was set to 1:3.

### 2.3. Study Questionnaire

The questionnaire included 32 questions in total and Likert scale graded answering options (Appendix A: Questionnaire). Children and their parents were asked to answer the questionnaire together. The questionnaire included the following three sections: Background data, GI symptoms, and Dietary effects on GI symptoms and daily life.

Background data were investigated by nine questions on growth, other diseases, medications, and allergies [18].

GI symptoms were assessed by both the Rintala Bowel Function Score (BFS) [19,20] and specific questions on the GI symptoms of constipation, abdominal pain, and bothersome gases [18]. The BFS is reported to have been validated for children aged 4–18 years, including, in total, seven questions generating a total score of 1–20 (20 = best function) [19,20]. A BFS of 18–20 suggests normal bowel function and ≤17 impaired bowel function [19,20]. Only answers for study participants aged 4–18 years who had answered all the BFS questions were included in the BFS analysis.

The third section was an assessment of the dietary effects on GI symptoms and daily life by using the Diet and Bowel Function questionnaire [13]. This instrument was developed for use by children aged 1–18 years old and their parents. It consists of 13 questions with graded (Likert scale of severity; Never to Always) or dichotomized (Yes/No) answering options. At the end of the questionnaire, there are questions about 90 food items that have been reported to induce GI symptoms in patients with GI disorders and which GI symptoms they cause (none/laxative/constipating/abdominal pain/gas symptoms/other) (Appendix A: Questionnaire).

### 2.4. Statistical Analysis

The statistical design was planned by a statistician. Only questionnaires with a minimum of 90% (29/32) questions answered were included in the study. Descriptive data were reported as *n* (%). Data on continuous variables did not fulfill assumptions for a normal distribution. They were therefore reported as the median (range). The non-parametrical Mann-Whitney U-test was used for continuous variables and for graded data on an ordinal scale. Binary categorical variables were analyzed by means of the Chi-square test. Statistical significance was set at *p* < 0.05. The program IBM SPSS Statistics Version 27 for MacOS was used.

HD was classified as either rectosigmoid aganglionosis, long-segment aganglionosis (extending past the rectosigmoid), or total colonic aganglionosis (TCA). Children with long-segment aganglionosis and TCA were grouped since the numbers of patients were few and no one had a longer extension of aganglionosis than in the colon. Answers to questions about constipation were dichotomized into No versus Yes (including experienced but no treatment, or treated with diet, medication, enemas, or other means). GI complaints were defined as constipation (Yes), abdominal pain (Often/Always) and/or bothersome gases (Often/Always).

### 2.5. Ethical Considerations

All methods were carried out in accordance with national guidelines and regulations. This study had received ethical approval from the Swedish Ethical Review Authority (registration number 2018/720). Participants received age-adapted oral and written information. Informed consents were obtained from both legal guardians for children younger than 15 years old, while adolescents aged 15 years or older gave their own consent.

## 3. Results

### 3.1. Background Information

In total, 71/85 (84%) children with HD and 265/300 (88%) controls fulfilled the inclusion criteria. This corresponded to a case-control ratio of 1:3.7. Figure 1 shows details of the study design and Table 1, background information.

Among children with HD, 58 (82%) had rectosigmoid aganglionosis, eight (11%) had long-segment colonic aganglionosis, and five (7%) had total colon aganglionosis (TCA), including a maximum resection of 10 cm of small bowel.

In univariate subgroup analyzes, neither aganglionic extension, age, or sex constituted risk factors for dietary GI effects or GI complaints (Appendix A: Diet and GI symptoms in HD subgroups. Appendix A: Bowel function according to BFS in children with HD with rectosigmoid vs. long-segment aganglionosis. Appendix A: GI symptoms (constipation, abdominal pain, bothersome gases) in children with HD with rectosigmoid vs. long-segment aganglionosis). Therefore, in the following analyzes, all children with HD were grouped.

### 3.2. Self-Reported GI Symptoms

Of the study participants aged 4–18 years old, the BFS was answered completely by 50 children with HD and 201 controls. Children with HD reported an inferior total score compared to controls (15 (4–20) vs. 20 (16–20), *p* < 0.001) A ‘Good bowel function’ (BFS 18–20) was reported by 15/50 (30%) children with HD, compared to 182/201 (90%) controls (*p* ≤ 0.001). Every single GI symptom listed in the BFS was reported inferiorly by children with HD (*p* ≤ 0.001 per item) compared to controls (Table 2).

The GI symptoms of constipation, abdominal pain, and/or bothersome gases were analyzed for children aged 1–18 years old with HD (*n* = 71) and for controls (*n* = 265) (Table 1). Overall, GI complaints (Often/Always) were reported by 72% of children with HD and by 29% of controls (*p* ≤ 0.001). For specific symptoms, abdominal pain (Often/Always) was reported by 20% of children with HD and 8% of controls (*p* = 0.004), while bothersome gases (Often/Always) were reported by 53% and 10% of the children, respectively (*p* ≤ 0.001).

Constipation was reported by 33% of children with HD vs. 19% of controls (*p* = 0.013) (Table 1). Among children reporting constipation, 5/23 (22%) with HD and 9/51 (18%) controls reported concomitant abdominal pain, and 11/23 (48%) and 13/51 (26%) reported bothersome gases. Current treatment for any GI symptoms was reported by 52/71 (73%) children with HD and 28/261 (11%) controls (*p* ≤ 0.001).

### 3.3. Dietary Effects on GI Symptoms

Dietary effects were reported more frequently by children with HD than by controls (77% vs. 52%, *p* ≤ 0.001). Children with HD also reported more frequently that they used dietary adjustments to improve bowel function (69% vs. 32%, *p* ≤ 0.001), both with regard to choosing and especially avoiding foods in order to improve GI symptoms. ‘Choosing’ was reported by 38% of children with HD vs. 16% of controls (*p* ≤ 0.001), and ‘avoiding’ by 62% vs. 18%, respectively (*p* ≤ 0.001). In sub-analysis including only children with GI complaints, these differences remained (Table 3).

### 3.4. Dietary Effects on Emotions and Daily Life

Social limitations as a result of dietary adjustments were reported more frequently by children with HD than by controls (20/71 [42%] vs. 22/265 [18%], *p* = 0.002). Moreover, concerns about dietary effects on GI symptoms differed between children and their parents: 26/71 children with HD (38%) vs. 60/265 controls (23%) (*p* = 0.003 for children’s views), and 62/71 parents of children with HD (89%) vs. 161/265 parents of control children (63%) (*p* ≤ 0.001 for parents’ views). Emotional effects as a result of dietary issues were reported to affect children with HD more frequently than controls (22/71 [32%] vs. 43/265 [17%], respectively, *p* = 0.004) and also their parents (36/71 parents of children with HD [51%] vs. 68/265 parents of controls [27%], *p* ≤ 0.001). Children with HD more frequently answered that they were interested in improving their knowledge about the effect of diet on GI symptoms compared to controls (43/71 [61%] vs. 55/265 [21%], respectively, *p* ≤ 0.001). In sub-analyzes including only children reporting GI complaints, these differences remained (Appendix A: Dietary Implications on Emotions and Social Life).

### 3.5. Effects of Food Items on GI Symptoms

More children with HD than controls reported the development of GI effects as a result of consuming any of the 18 vegetables enquired about in the questionnaire: 36/71 (51%) vs. 84/265 (32%), respectively, *p* = 0.001, and by dairy products: 29/71 (41%) vs. 53/265 (20%), respectively, *p* ≤ 0.001. GI effects as a result of consuming any of the 18 fruits inquired about did not differ significantly between children with HD and controls: 30/71 (42%) vs. 92/265 (35%), respectively, *p* = 0.056.

In children with GI complaints, children with HD more frequently than healthy controls reported GI effects of vegetables: 31/51 (61%) vs. 24/77 (31%), respectively (*p* = 0.001), while GI effects of dairy products and fruits did not differ between the groups (dairy effects: 23/51 [45%] vs. 24/77 [31%], respectively, *p* = 0.100, and fruits: 25/51 [49%] vs. 31/77 [40%], respectively, *p* = 0.118).

Among all the 90 food items enquired about, the median number of items reported to induce GI symptoms was significantly higher for all children with HD than healthy controls (7 (0–66) vs. 2 (0–34), *p* ≤ 0.001) and for children with GI complaints (9 (0–59) for children with HD vs. 3 (0–34) for controls, *p* = 0.012). Specific GI effects as a result of consuming selected types of food items are displayed in Figure 2.

The types of food items reported to induce GI symptoms and the frequency of children reporting on each type differed between children with HD and controls (Appendix A: Food items and GI symptoms). The one food item reported most frequently to induce GI symptoms in children with HD was milk, reported by 26/71 (37%) vs. 37/265 (14%) controls, *p* ≤ 0.001, then banana 24/71 (34%) vs. 27/265 (10%) controls, *p* ≤ 0.001, and corn 24/71 (34%) vs. 13/265 (5%) controls, *p* ≤ 0.001. Exactly the same ranking (milk, banana, and corn) was reported by children with GI complaints (Appendix A). For controls, the most frequently reported food item to induce GI symptoms was beans, but the frequency was still less than that reported by children with HD (63/265 [24%] for controls vs. 21/71 [30%] for children with HD, *p* = 0.013). For controls, the next most frequently reported food items were plums (55/265 [16%], vs. 11/71 [16%] for children with HD, *p* = 0.318), and cream, reported by 38/265 (14%) controls, which was still less than in children with HD (23/71 [32%], *p* ≤ 0.001) (Appendix A). The same ranking of causative food items (beans, cream, and plums) was also reported by controls with GI complaints (Appendix A).

## 4. Discussion

Children with HD reported more frequently than controls that diet induced GI symptoms and made dietary adjustments accordingly, which influenced their daily lives. Specific food items inducing GI symptoms were also cited more frequently by children with HD than by controls, and these differed between the two groups. The differences persisted when comparing only children reporting GI complaints. Similar explicit dietary differences comparing children with HD and controls have never been presented before. The results reveal that diet is a stronger extrinsic factor for GI symptoms in children with HD than is the case for healthy children [14]. These quantitative measures not only confirm but also explain the results of the focus group study of parents of children with HD [12].

In this study, 77% of children with HD reported that diet induced GI symptoms. This proportion was higher than that reported previously, where 58–64% of patients with HD reported dietary-induced GI symptoms [9,10]. Additionally, in our study, children with HD reported more frequent effects of diet on their GI symptoms compared to controls. The differences remained, even when sub-analyzing only children with GI complaints.

A well-known GI symptom in children with HD is let-out obstruction [1,2,3,4]. This symptom could easily be misinterpreted and treated incorrectly as constipation. A high-fiber diet, such as one rich in fruits and vegetables, is frequently recommended to treat constipation [21], although evidence for its efficacy is lacking [8,22]. In our study, the consumption of fruits and vegetables mainly induced pain and gas complaints, while laxative effects were reported only rarely by children with HD. Therefore, consuming a high-fiber diet could become burdensome for them. Again, these results confirm the main issues raised by parents of children with HD within focus group discussions [12]. Accordingly, it seems that dietary recommendations to treat constipation, which are recommended for otherwise healthy children [5,6,7,8], could even be contraindicated in children with HD.

Most parents of children with HD (89%) reported concerns about dietary effects on their child’s GI complaints, and nearly 40% of children with HD reported an influence of dietary adjustments on daily life. The results confirm our previous findings, revealing frequent parental worries about worsening their child’s GI symptoms by giving them the wrong foods [12]. Similar psychosocial concerns regarding diet have been reported in families of children with food allergies [23,24]. This psychological stress highlights the importance of adopting a multi-professional approach to management, encompassing social workers, psychologists, and dieticians early on, in order to diminish any psychosocial suffering.

The main strength of our study was to address diet’s role in GI complaints in HD, which has been shown to be a highly ranked research topic by patient organizations [12,25,26]. The study questionnaire used was developed in collaboration with the national patient association for HD, thus ensuring its relevance and end-user feasibility [12,13,17]. Involving patients in the design of studies has been shown to generate higher-quality and more relevant research [27]. Another strength of the study was that not just general food groups were investigated but also specific food items, which revealed more detailed knowledge about their dietary effects, which is essential for future recommendations.

One limitation of the study was the cross-sectional design, implying that neither changes over time nor causalities were investigated. Moreover, the large number of food items specified in the questionnaire could be criticized as a result of the risk of answer exhaustion. However, on the other hand, very few participants added other food items in the free text section, suggesting that most of the food items relevant to them had been covered in the questionnaire. Regarding food allergies, it cannot be excluded that the few IgE-mediated food allergies reported affected the results, but the probability of bias was considered to be small. This is since the frequencies were similar in the groups and also since effects on GI symptoms caused by milk allergy are reported to occur only in up to 30% of cases [28]. A selection bias toward children with GI complaints or food-focused families could not be excluded, although the dropout rate of children with HD (16%) was low. The absence of significant differences in GI symptom severity between children with different extensions of aganglionosis may be attributed to the limited number of children with long-segment aganglionosis. The weak power could, hypothetically, in turn lead to the absence of differences regarding dietary effects on GI symptoms between subgroups of children with HD. In order to definitively determine if dietary GI effects are/are not associated with the severity of disease, i.e., extension of aganglionosis, larger multicenter studies are needed. Another source of uncertainty is the difficulty in distinguishing between parents’ and children’s reports since they were requested to answer together. In both groups, twice as many parents as children reported worries about dietary effects on GI symptoms. This might be due to a misunderstanding of the question, which was intended for only the children. The generalizability of our results could also be questioned since the questionnaire used was only provided in one language (Swedish) and the food items listed in it reflected a typical Nordic diet. In order to confirm results in a wider international context, linguistic and cultural adaptations to the questionnaire are required.

The study’s results support the inclusion of dietary changes in bowel management programs within HD guidelines [1], and the results could guide healthcare workers in optimizing HD-specific dietary recommendations. Applying the results to clinical use, it would seem that advice is better directed towards selecting/avoiding specific food items, e.g., beans, instead of food group categories, e.g., ‘fruits’ or ‘vegetables.’ This is because specific food items within a food group affect GI symptoms differently, especially in children with HD. Furthermore, because of the reported high incidence of side effects (gas and stomach pain) associated with adopting a high-fiber diet (which is usually recommended for children with constipation), such a diet might be contraindicated in children with HD suffering from outlet obstruction. Taking into account the very limited positive effect of diet on GI symptoms in healthy children [8,14], expectations for improving HD symptoms by changing diet could be lowered. Still, according to our study, diet appears to impact GI symptoms in children with HD to a greater extent than in healthy children; however, even so, there is a need for great caution when advising any dietary modifications for children with HD. In order to clarify the true role of diet in the management of HD and before any evidence-based recommendations can be given, dietary interventional studies must be carried out. Nevertheless, the results of this study serve to increase our knowledge about the effects of diet on GI complaints in children with HD, and they support the planning of future interventional studies.

## 5. Conclusions

Children with HD report more frequently than controls that diet induces GI symptoms and make dietary adjustments that influence their daily lives. Children with HD also report that more food items induce GI symptoms and that their GI effects are different and more frequent compared to controls. These dietary differences are also evident in children with GI complaints. The results strongly suggest a need for disease-specific dietary advice for children with HD, and for further studies.

## Figures and Tables

**Figure 1 children-10-01543-f001:**
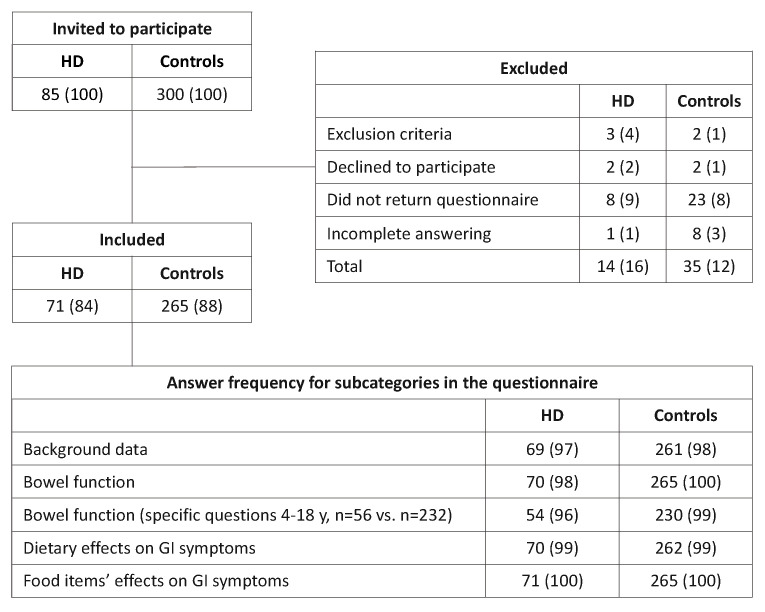
Flowchart of participants (children with Hirschsprung’s disease and controls 1–18 years old) included and excluded in the study and answer frequency for each category in the questionnaire. Values are *n* (%) of children. HD = Hirschsprung’s disease. GI = Gastrointestinal.

**Figure 2 children-10-01543-f002:**
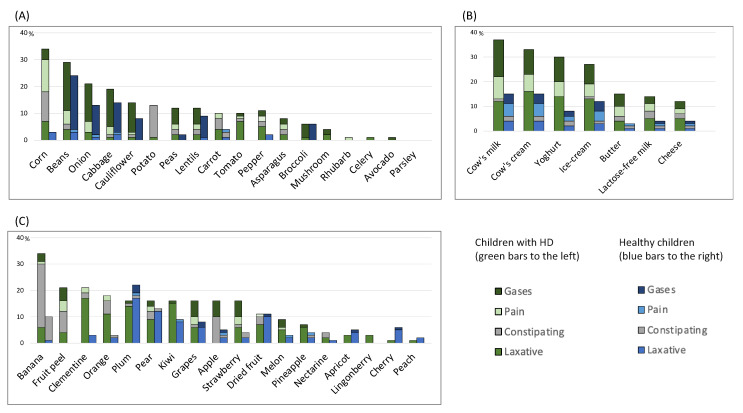
Comparison of frequencies of (**A**) vegetables (**B**) dairy and (**C**) fruits inducing gastrointestinal symptoms, and their specific gastrointestinal effect, as reported by children with Hirschsprung’s disease (HD) (*n* = 71) (green bars to the left) and healthy children (*n* = 265) (blue bars to the right) using the Diet and Bowel Function questionnaire.

**Table 1 children-10-01543-t001:** Participant characteristics and gastrointestinal symptoms in children aged 1–18 years old with and without Hirschsprung’s disease using the Diet and Bowel Function questionnaire. Values are *n* (%) of children.

Participant Characteristics	HD *n* = 71	Controls *n* = 265	
Age, years	Median 6 (range 1–17)	Median 9 (range 1–18)	
Boys/Girls	56 (79)/15 (21)	137 (52)/128 (48)	
Concomitant allergy *	6 (8)	30 (11)	
Eating a special diet (e.g., vegetarian or vegan food)	16 (23)	33 (13)	
Current treatment for GI symptoms	52 (73)	28 (11)	
Type of treatment			
Adjusted diet	14 (20)	10 (4)	
Medicine	15 (21)	16 (6)	
Enemas	14 (20)	1 (0)	
Other treatment	2 (3)	1 (0)	
Reason for treatment			
Constipation	30 (42)	17 (6)	
Diarrhea	8 (11)	2 (1)	
Stomach pain	2 (3)	2 (1)	
Other reason	8 (11)	1 (0)	
Self-reported GI symptoms			
			*p*-value
Constipation			
No constipation	46 (67)	212 (81)	*p* = 0.013
Constipation without treatment	1 (1)	30 (11)	
Manageable with diet	2 (3)	5 (2)	
Manageable with medicine	16 (23)	15 (6)	
Not manageable with either diet or medicine (enemas)	4 (6)	1 (0)	
Missing	2 (3)	2 (1)	
Abdominal pain			
Never	22 (31)	84 (32)	*p* = 0.004
Sometimes (at most once a week)	35 (49)	159 (60)	
Often	14 (20)	21 (8)	
Always	0 (0)	0 (0)	
Missing	0 (0)	1 (0)	
Bothersome gases			
Never	6 (9)	138 (52)	*p* ≤ 0.001
Sometimes (at most once a week)	28 (39)	99 (37)	
Often	31 (44)	25 (9)	
Always	6 (9)	3 (1)	
Missing	0 (0)	0 (0)	
GI symptoms: constipation, pain and/or gases			
No/Seldom GI symptoms	20 (28)	188 (71)	*p* ≤ 0.001
GI complaints (Often/Always)	51 (72)	77 (29)	

* HD: celiac disease *n* = 2, dietary fiber intolerance *n* = 1, allergy to egg *n* = 1, caviar *n* = 1, milk protein *n* = 1, other *n* = 2; Controls: allergy to milk protein *n* = 8, lactose intolerance *n* = 7, nuts *n* = 4, egg *n* = 3, fish *n* = 3, other (apple, bread, clove, other protein allergy, meat) *n* = 5; HD = Hirschsprung’s disease; GI = gastrointestinal.

**Table 2 children-10-01543-t002:** Bowel function in children aged 4–18 years old with and without Hirschsprung’s disease according to the Rintala Bowel Function Score (BFS). Numbers are (percentages) or medians (ranges).

	Score	HD *n* = 50	Controls *n* = 201	*p*-Value
Ability to hold back defecation				
Always	3	27 (54)	192 (96)	<0.001 ^1^
Problems < 1/week	2	14 (28)	6 (3)	
Problems > 1/week	1	4 (8)	3 (1)	
Never	0	5 (10)	0 (0)	
Feels the urge to defecate				
Always	3	27 (54)	182 (91)	<0.001 ^1^
Often	2	10 (20)	17 (8)	
Sometimes	1	11 (22)	2 (1)	
Never	0	2 (4)	0 (0)	
Frequency of defecation				
Every other day—twice a day	2	29 (58)	162 (81)	<0.001 ^1^
More often	1	21 (42)	20 (10)	
Less often	1	0 (0.0)	19 (9)	
Soiling				
Never	3	12 (24)	159 (79)	<0.001 ^1^
Sometimes (<1/week)	2	15 (30)	38 (19)	
Often	1	17 (34)	4 (2)	
Always	0	6 (12)	0 (0)	
Accidents				
Never	3	33 (66)	197 (98)	<0.001 ^1^
Sometimes (<1/week)	2	12 (24)	3 (2)	
Often (>1 week)	1	2 (4)	1 (0)	
Daily	0	3 (6)	0 (0)	
Constipation				
No constipation	3	33 (66)	186 (93)	<0.001 ^1^
Manageable with diet	2	2 (4)	4 (2)	
Manageable with medicine	1	12 (24)	10 (5)	
Not manageable with either diet or medicine	0	3 (6)	1 (0)	
Social problems				
No social problems	3	28 (56)	181 (90)	<0.001 ^1^
Sometimes	2	13 (26)	19 (10)	
Problems causing restrictions in social life	1	8 (16)	1 (0)	
Severe social and/or psychological problems	0	1 (2)	0 (0)	
Total Bowel Function Score				
BFS	1–20	15 (4–20)	20 (16–20)	<0.001 ^1^

^1^ Mann-Whitney U-Tests, two-tailed. HD = Hirschsprung’s disease.

**Table 3 children-10-01543-t003:** Dietary effects on gastrointestinal (GI) symptoms in children aged 1–18 years old with and without Hirschsprung’s disease (HD) using the Diet and Bowel Function questionnaire. GI complaints were defined as constipation (Yes), abdominal pain (Often/Always), and/or bothersome gases (Often/Always). Values are *n* (%) of children.

	All Children *n* = 336			Children with GI Complaints *n* = 128		
	HD *n* = 71	Control *n* = 265	*p*-Value	HD *n* = 51	Control *n* = 77	*p*-Value
Diet induces GI symptoms						
No	16 (23)	127(48)	<0.001 ^1^	10 (20)	25 (33)	0.004 ^1^
Sometimes	29 (41)	116 (44)		18 (35)	35 (46)	
Often	13 (18)	18 (7)		12 (24)	14 (18)	
Always	13 (18)	3 (1)		11 (22)	3 (4)	
Missing	0 (0)	1 (0)		0 (0)	0 (0)	
Eating habits induce GI symptoms						
No effects	38 (55)	164 (62)	0.126 ^1^	23 (47)	44 (58)	0.130 ^1^
Sometimes	21 (30)	86 (33)		16 (33)	24 (32)	
Often	5 (7)	10 (4)		5 (10)	6 (8)	
Always	5 (7)	4 (2)		5 (10)	2 (3)	
Missing	2 (3)	1 (0)		2 (4)	1 (1)	
No adjustments	22 (31)	181 (68)	<0.001 ^1^	12 (24)	33 (43)	0.003 ^1^
Sometimes	23 (32)	60 (23)		18 (35)	28 (36)	
Often	11 (16)	14 (5)		9 (18)	11 (14)	
Always	15 (21)	10 (4)		12 (24)	5 (7)	
Missing	0 (0)	0 (0)		0 (0)	0 (0)	
Yes	27 (38)	41 (16)	<0.001 ^2^	20 (39)	23 (30)	0.273 ^2^
No	44 (62)	223 (85)		31 (61)	54 (70)	
Missing	0 (0)	1 (0)		0 (0)	0 (0)	
Yes	44 (62)	47 (18)	<0.001 ^2^	32 (63)	23 (30)	<0.001 ^2^
No	27 (38)	217 (82)		19 (37)	53 (70)	
Missing	0 (0)	1 (0)		0 (0)	0 (0)	

^1^ Mann-Whitney U-Test; ^2^ Chi-square test.

## Data Availability

The data presented in this study are available on request from the corresponding author. The data are not publicly available due to maintenance of confidentiality and protection of each participant’s personally identifiable information.

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
