# Peer review of "Children with Hirschsprung’s Disease Report Dietary Effects on Gastrointestinal Complaints More Frequently than Controls"

_children, 2023, doi:10.3390/children10091543_

Round 1

Reviewer 1 Report

1.       Introduction- it is well written and states the addresses in article issue well. You may consider mentioning that part of the cited research was yours (one can get to the conclusion only after reading references).

2.       Materials and Methods- the study setting and study population are described properly.  

a)       I have some doubts about the subgroup with self-reported food allergies and/or intolerances. You write that you only included those without medication, but as you well know most of those conditions are treated with diet, so you can assume that most of that subpopulation will report some sort of GI symptoms related to food. Since this subgroup only consist of 8% an 11%  of study population (HD/controls) please consider excluding this subgroup from your analysis- would this influence your outcome?

b)      HD- group- vast majority (82%) had rectosigmoid aganglionosis- which is probably why you did not detect significant differences between HD subgroups, nevertheless it is hard to believe that the aganglionic extension has no influence on GI symptoms severity. Would you get the same results were the subgroup in HD population be even? (where rectosigmoid aganglionosis would be maximum 50% of the HD group)

I think it is worth mentioning it in the discussion among studies limitations.

3.       Results-

-          since the study was designed that way that the questions were send to the patients I wonder why over 80% of HD group had rectosigmopid aganglionosis, was it calculated on purpose or were the patients from other subgroup excluded due to other co-morbidities?

-          you may consider putting Table S2 in the main article not supplement.

-          I wonder what is the difference in table 2 between adjusting diet and together: choosing and avoiding food, are you sure that patients understood the difference?

-          In section 3.4- what stroke mi was the difference of percentage of children themselves and parents in regard to concerns about dietary effect on GI symptoms (parents worry twice as much in both HD and control group). Do you thing this may be a bias to your study results? I think it is worth mentioning in the discussion section,

-          In section 3.5 you mention milk as the one item reported most frequently to have effect on GI symptoms. Did you check how much from the subgroup with allergies or intolerances were in those selecting this product? Since it is commonly not tolerated product.  It is worth mentioning in the discussion section.

Author Response

Thank you very much for your comments and suggestions. Please see the attachment. 

Reviewer 2 Report

The manuscript provides a preliminary overview of the study's findings related to GI symptoms and dietary effects in children with HD compared to controls. While the study design, data interpretation and discussion seem well-structured, there are areas for improvement in terms of methodology. Further clarification and expansion on these aspects would enhance the scientific rigor and overall quality of the manuscript:

Given that HD extension encompasses distinct GI potential consequences, such as total colonic aganglionosis or sigmoid HD, the inclusion of additional details regarding various forms of HD should be considered. It might be beneficial to incorporate a subgroup analysis of GI symptoms based on these different forms. In the event that the authors are unable to furnish this information, it could introduce a potential bias, which should be acknowledged as a study limitation within the discussion section.

Author Response

(The authors gave the same response as above.)

Reviewer 3 Report

Author needs to address the following comments.

The author suggested writing the full address of the department in the study setting.

Figure 2, author needs to label the graph properly to identify the children with HD and Healthy children.

Figure 2B – what kind of cream used?

Figure 2A – Potato, mushroom, Rhubarb, celery, avocado, parsley 2C – Fruit peel, Lingonberry - shows only one bar?

Author Response

Thank you for your comments and suggestions. Please see the attachment.
